# Lung Extracellular Matrix Hydrogels-Derived Vesicles Contribute to Epithelial Lung Repair

**DOI:** 10.3390/polym14224907

**Published:** 2022-11-14

**Authors:** Anna Ulldemolins, Alicia Jurado, Carolina Herranz-Diez, Núria Gavara, Jorge Otero, Ramon Farré, Isaac Almendros

**Affiliations:** 1Unitat de Biofísica i Bioenginyeria, Facultat de Medicina i Ciències de la Salut, Universitat de Barcelona, 08036 Barcelona, Spain; 2The Institute for Bioengineering of Catalonia (IBEC), The Barcelona Institute of Science and Technology (BIST), 08028 Barcelona, Spain; 3CIBER de Enfermedades Respiratorias, 28029 Madrid, Spain; 4Institut d’Investigacions Biomèdiques August Pi i Sunyer, 08036 Barcelona, Spain

**Keywords:** extracellular matrix, hydrogel, mesenchymal stem cells, extracellular vesicles, lung epithelial cells, lung repair

## Abstract

The use of physiomimetic decellularized extracellular matrix-derived hydrogels is attracting interest since they can modulate the therapeutic capacity of numerous cell types, including mesenchymal stromal cells (MSCs). Remarkably, extracellular vesicles (EVs) derived from MSCs display similar functions as their parental cells, mitigating tissue damage in lung diseases. However, recent data have shown that ECM-derived hydrogels could release other resident vesicles similar to EVs. Here, we aim to better understand the contribution of EVs and ECM-vesicles released from MSCs and/or lung-derived hydrogel (L-HG) in lung repair by using an in vitro lung injury model. L-HG derived-vesicles and MSCs EVs cultured either in L-HG or conventional plates were isolated and characterized. The therapeutic capacity of vesicles obtained from each experimental condition was tested by using an alveolar epithelial wound-healing assay. The number of ECM-vesicles released from acellular L-HG was 10-fold greater than EVs from conventional MSCs cell culture revealing that L-HG is an important source of bioactive vesicles. MSCs-derived EVs and L-HG vesicles have similar therapeutic capacity in lung repair. However, when wound closure rate was normalized by total proteins, the MSCs-derived EVs shows higher therapeutic potential to those released by L-HG. The EVs released from L-HG must be considered when HG is used as substrate for cell culture and EVs isolation.

## 1. Introduction

The development and use of extracellular matrix (ECM)-derived hydrogels is attracting interest in cell therapy. The ECM is one of the most important cell niche components since it provides structural support for cells and is also critical in developmental organogenesis, homeostasis, and injury-repair responses. In particular, the physical signals exerted by the ECM composition, topography, and rigidity are translated to the cells via mechanotransduction which has been found to be crucial in regulating stem cell fate [1]. In contrast to traditional cultures, the biomechanical properties of hydrogels can be modulated to provide a physiomimetic 3D environment reproducing physiological and disease conditions in vitro [2,3,4,5]. The stemness and differentiation characteristics of mesenchymal stromal cells (MSCs) in decellularized ECM hydrogels are significant for stem cell therapy and for designing new treatments [3,4]. Among the different opportunities and challenges, ECM-derived hydrogels are biomechanically tunable and have been recently used as MSCs delivery systems [6] and bioinks for 3D bioprinting [5] opening new possibilities for tissue engineering.

Regarding lung diseases, the most severe manifestation of acute lung injury is acute respiratory distress syndrome (ARDS), a hypoxemic respiratory failure characterized by severe impairment in gas exchange and lung mechanics with a high case fatality rate since there is no effective treatment [7]. From pre-clinical studies, MSCs seem to be effective in ameliorating lung permeability, modulating inflammatory mediators and facilitating lung repair [8]. However, clinical trials to date have not provided strong evidence for MSC efficacy [9,10]. Potential limitations are the safety of these cells, their obtention procedure, and their viability after transplantation. Interestingly, MSCs cultured on decellularized lung scaffolds [11] and native lung-derived hydrogels [3] increased their therapeutic potential compared to conventional cell cultures. In a recent study, hydrogel-encapsulated MSCs could further alleviate acute lung injury, increasing the expression of several growth factors and interleukin-10 [12]. In addition, hydrogel-encapsulated MSCs showed better cell survival and could increase their engraftment in the injured tissue [13]. Thus, biomechanical preconditioning of cultured MSCs in physiomimetic hydrogels is a promising strategy to improve cell therapy in future clinical trials.

During the last few years, several studies have emerged showing that the therapeutic effects of MSCs are largely mediated by paracrine factors, which are transported within extracellular vesicles (EVs). Increasing evidence suggests that MSCs-derived EVs might represent a novel cell-free cell therapy with compelling advantages, compared with using parent MSCs, such as no risk of tumor formation and even lower immunogenicity. Regarding ARDS, MSC-derived EVs seem to actively participate during the normal recovery process [14]. Moreover, several groups have revealed a therapeutic advantage when administering MSC-derived EVs by intratracheal infusion or intravascular route. Intratracheal instillation of MSC-derived EVs in *Escherichia coli* lipopolysaccharide (LPS)-induced lung injury improved pulmonary edema, inflammation, and the integrity of the alveolar-capillary barrier [15,16]. However, how the physicochemical preconditioning of MSC cultured on physiomimetic hydrogels can modulate the therapeutic capacity of their secreted EVs is still largely unknown.

In contrast to conventional culturing, isolation and characterization of EVs secreted by MSCs cultured in decellularized ECM-derived hydrogels may pose a challenge since ECM bioscaffolds can also release bioactive vesicles with a structure (round vesicles) and size (50 to 400 nm) similar to EVs released by cells [17]. Thus, these ECM-vesicles represent another source of bioactive vesicles which can be mixed with the EVs released by the MSCs cultured on ECM derived-hydrogels. In addition, it has been reported that ECM-vesicles released from ECM fibrils by digestion contain miRNAs which have been associated with cellular development, proliferation, survival, migration, and cell cycle activity [17]. Regarding lung diseases, ECM infusion was able to improve cell survival and alveolar morphology and reduced hyperoxia-induced apoptosis and oxidative damage in a murine model of acute lung injury [18]. Moreover, recent studies have shown that some ECM components released from the hydrogel could provide some beneficial functions including antibacterial activity [19], and cell proliferation and chemotaxis [20]. However, the mechanisms involved are unknown. Herein, we hypothesize that ECM-bound vesicles could be an important component released from the hydrogels facilitating lung repair synergistically with those EVs released from MSCs. To this end, we tested the potential therapeutic effects of both EVs released from MSCs and/or ECM-vesicles from lung-derived hydrogel (L-HG) in a wound-healing experimental model of lung repair. The EVs and ECM-vesicles were characterized and quantified, and their contribution to repairing an alveolar epithelial cell monolayer was tested.

## 2. Materials and Methods

### 2.1. Preparation of the Lung ECM-Derived Hydrogels

Porcine lungs were obtained from a local slaughterhouse and decellularized as previously described [5]. Lungs were perfused through the trachea and the vasculature with 0.1% Triton X-100, sodium deoxycholate, DNase and 1 M sodium chloride, with intermediate perfusion with distilled water and PBS for rinsing purposes. To assess the effectiveness of decellularization, total genomic DNA was isolated using the PureLink Genomic DNA kit (ThermoScientific, Waltham, MA, USA) from native and decellularized scaffolds following the manufacturer’s instructions. The total amount of DNA was quantified using spectrophotometry and normalized to the sample tissue dry weight. The amount of DNA was 16.26 ± 4.24 ng per mg of dry tissue which is below the accepted threshold of 50 ng/mg for successful decellularization (Appendix A) [21].

Decellularized ECM were drained of excess water, freeze-dried in pieces, and lyophilized (Telstar Lyoquest55 Plus, Terrassa, Spain). Afterward, the sample was pulverized into micron-sized particles at −180 °C by using a cryogenic mill (6755, SPEX, Metuchen, NJ, USA) for 5 min at maximum speed. The resulting powder was digested at 20 mg/mL concentration in a 0.01 M HCl solution with pepsin from porcine gastric mucosa (1:10 concentration) under magnetic stirring at room temperature for 16 h. The resulting (pregel) solution was then pH-adjusted to 7.4 (±0.4) by using 0.1 M NaOH and PBS 10X and frozen at −80 °C for subsequent use. The biocompatibility of the L-HG has been confirmed as well as the attachment and growth of the cells within the lung ECM construct [4,5].

### 2.2. Rat Bone Marrow Mesenchymal Stromal Cells Isolation

Primary rat bone marrow-derived MSCs were isolated following an adapted protocol from [22] and approved by the Ethical Committee for Animal Research of the University of Barcelona. Bone marrow from the femurs and tibias of Sprague-Dawley rats (250 g) was flushed with a 19 G needle and syringe with prewarmed supplemented DMEM (Gibco) medium. The whole mesh was collected and disaggregated. Subsequently, cells were centrifuged at 350× *g* for 5 min and cultured on conventional plastic vessel T-75 (Techno Plastic Products AG, Trasadingen, Switzerland) with αMEM supplemented with 10% FBS and incubated with 5% CO_2_ balanced-air incubator at 37 °C. After three days, the medium was replaced to discard all non-adhered cells and trypsinized for 5 min with TrypLE express trypsin (Gibco) before they reached confluence. MSCs were phenotypically characterized by flow cytometry. Fluorescein isothiocyanate (FITC)- or phycoerythrin (PE)-conjugated monoclonal antibodies specific for CD29, CD44H, CD45, CD11b/c, and CD90 (BioLegend, San Diego, CA, USA) were used.

### 2.3. Extracellular Vesicles Isolation

Vesicles were isolated and quantified from L-HG alone, and MSCs supernatants cultured on L-HG or on conventional plastic plates. Briefly, 10 mL of medium was collected and centrifuged at 2000× *g* for 30 min, and 5 mL of Total Exosomes Isolation Reagent (Life Technologies) was added. The mixtures were incubated overnight at 4 °C, followed by centrifugation at 10,000× *g* for 1 h. The pellets were suspended in 1 mL of PBS 1X and stored at –80 °C for further use. The concentration and size of EVs were analyzed by Nanoparticle Tracking Analysis (NTA).

### 2.4. F-Actin Staining

After EVs’ isolation, cells were fixed with PFA 4% for 15 min, permeabilized with 0.1% Triton X-100, blocked with a 10% FBS solution and counter-stained for nuclei (NucBlue, ThermoScientific, Waltham, MA, USA) and F-actin (phalloidin, Thermoscientific, Waltham, MA, USA), and finally imaged with the Nikon D-Eclipse Ci confocal microscope ×20 Plan Apo immersion oil objective (Nikon, Minato ku, Japan).

### 2.5. Scanning Electron Microscopy

The hydrogel was freshly prepared and fixed in glutaraldehyde 2.5% in phosphate buffer (PB) 0.1 M, pH 7.4 for 24 h. After the fixation, it was washed ×4 with PB 0.1 M and treated with osmium tetraoxide 1% in PB at 4 °C for 90 min. Then, the sample was cleaned with ultrapure water until no yellow color was observed in the sample and water. The sample was cleaned with ethanol 50% followed by ethanol 70% and kept at 4 °C overnight. Then, it was dried by using serial dilutions of ethanol from 80% to 100%. Finally, the sample was critical point dried (Autosamdri-815 critical point dryer, Tousimis, Rockville, MD, USA), gold coated, and mounted using conductive adhesive tabs (TED PELLA, Redding, CA, USA). Imaging was performed by using scanning electron microscopy (SEM) (JSM-6510, JEOL, Tokyo, Japan) at 15 kV.

### 2.6. Transmission Electron Microscopy

A 30 µL measure of the sample was placed in a clean parafilm piece and on top of the drop, a 400 copper mesh grid with formvar membrane and incubated for 25 min. The face of the grid that was in contact with the sample was then placed on top of a 30 µL staining agent (uranyl acetate 2%) drop and kept in contact for 30 s. Subsequently, samples were allowed to dry on a petri dish with filter paper for at least 1 h before being observed with transmission electron microscopy (TEM). The images were performed using a J1010 (Jeol) coupled with Orius CCD camera (Gatan) on the TEM-SEM Electron Microscopy Unit from Scientific and Technological Centers (CCiTUB) at the University of Barcelona.

### 2.7. Protein Quantification

First, to investigate the delivery of total protein and proteins from ECM-vesicles from acellular L-HG to the medium, the isolation protocol and quantification were measured in supernatants on different days. To this end, L-HG were washed with PBS 1× for 1 week. Every day, the PBS 1× was removed, and αMEM basal medium (without FBS) was added for 48 h to replicate cell culture conditions. After that, the EVs isolation protocol was applied. In parallel, L-HG were cultured for 1 week in PBS 1×, and the supernatant was collected daily. The protein content either from de L-HG supernatant or after the EVs isolation was assessed by BCA Protein kit (23225, Thermo Scientific™, Waltham, MA, USA).

These results showed that the delivery of vesicles from hydrogels reached a plateau after 3 days. Thus, experiments with MSCs embedded in L-HG were carried out after 3 days of L-HG washes in parallel to acellular L-HGs and MSCs in plastic cultures.

### 2.8. Wound Healing Assay

Rat lung epithelial cells (RLE) (CRL-2300, ATCC, Manassas, VA, USA) were used. The wound closure on RLE after the exposure of EVs and/or ECM-vesicles was assessed as previously described [23]. Briefly, a density of 2·10^5^ cells/cm^2^ were seeded until cell reached confluency. Then, the epithelial cell monolayer was scratch-wounded using a sterile 200-µL pipette tip (Eppendorf, Hamburg, Germany), and cell debris was removed by washing with PBS 1×. Subsequently, PBS was discarded and replaced by a free-serum medium with a 1/10 dilution of the vesicles isolated from the different culture conditions. For any given condition, a parallel control group (PBS 1X instead of EVs) was included to further normalize the wound healing rate. For each experimental group, the wound healing measurements were carried out on 3 different days in duplicate (*n* = 5/experimental group).

The wound area was measured immediately after the scratch performance (0 h) and at the end of the experiment (24 h). Phase contrast images were recorded with an inverted microscope (Eclipse Ti, Nikon Instruments, Amsterdam, The Netherlands) equipped with a camera (C9100, Hamamatsu Photonics K.K., Hamamatsu, Japan) and a 10 × objective. Wound closure was assessed by comparing each epithelial wound’s initial and final area with ImageJ 2.1.0/1.53c software, (NIH, Bethesda, MD, USA). Briefly, using the freehand selection tool by a researcher unaware of the experiment cell group, the edges were marked in all images, and the areas were calculated. The wound closure was computed as the percentage change between the two time points.

### 2.9. Statistical Analysis

Data are presented as mean ± standard error (SE). Comparisons between groups were made by one-way analysis of variance (ANOVA), except for the release of proteins at different time points where repeated two-way ANOVA was done. Student–Newman–Keuls post hoc test was used for multiple comparisons. Differences were considered significant for *p* < 0.05. Statistical analyses were performed with SigmaPlot (v13.0 Systat Software, San Jose, CA, USA).

## 3. Results

### 3.1. Acellular Lung Hydrogel Is an Important Source of Extracellular Vesicles

SEM images of the L-HG hydrogels showed the fibrillary architecture of the scaffold (Appendix A). MSCs cultured on plastic and on L-HG showed a spread morphology indicative of their normal attachment (Appendix A). Actin filaments were distributed in a similar way in both conditions and with the characteristic spindle-like morphology.

As expected, L-HG releases a large amount of ECM-bonded proteins to the medium (1250 µg/mL) that decreases after consecutive daily washes (61.9 µg/mL after 7 days) (Figure 1). The proportion of proteins corresponding to those encapsulated in vesicles was approximately a 1/6 ratio with respect to the total protein release (227 µg/mL). Similarly, the proteins from ECM-bonded vesicles were reduced after daily washes, reaching a plateau on the third day (19 µg/mL) (Figure 1).

Taking into consideration these data, we selected acellular L-HG after 3-day washing to use for MSCs culture in order to minimize the effects of L-HG-derived EVs on those secreted by the own cells. Interestingly, TEM images showed that after conventional EVs isolation protocol, there are isolated vesicles but also ECM-bound vesicles (Figure 2A). Although the amount of proteins released by the L-HG was reduced, NTA analysis revealed that there was still an important secretion of nanoparticles with similar physical characteristics to that released by MSCs (Figure 2). Vesicles isolated from MSCs cultured on conventional plastic and ECM-vesicles isolated from acellular L-HG presented a very similar NTA profile with a diameter of approximately 180 nm (Figure 2B). Therefore, the differential identification of EVs and ECM-vesicles released from MSCs and L-HG, respectively, was not possible when MSCs are cultured on L-HG.

The protein content from isolated EVs and ECM-vesicles was also measured to better estimate the total amount of EVs released in each condition. In accordance with NTA analysis, the protein amount obtained from isolated vesicles was 3-fold higher in those isolated from acellular L-HG (309 µg/mL, *p* < 0.01) and from L-HG with MSCs (363 µg/mL, *p* < 0.01) with respect to that obtained in MSCs cultured in plastic (134 µg/mL) (Figure 2C).

### 3.2. Woung Healing Is Enhanced by MSCs and L-HG-Derived EVs

The percentage of wound closure was increased when EVs and/or ECM-vesicles were applied (Figure 3). A group of ECM-vesicles isolated from L-HG without washes was also included in this assay. The greater closure was achieved when L-HG-derived vesicles were present and specifically for those obtained from non-washed L-HG (12.9% vs. 7.1% controls, *p* < 0.05) (Figure 3B). Considering that the number of vesicles released from MSCs was lower than that obtained from any other L-HGs condition when wound closure was normalized by the total amount of protein applied (Appendix A), the EVs derived from MSCs were the most efficient for wound healing (Figure 3C).

## 4. Discussion

Studies using physiomimetic ECM-derived hydrogels have recently emerged as a promising in vitro model that replicates many features of the cell environment in native healthy tissues or pathological conditions. Regarding cell therapy, hydrogels provide clear advantages to conventional culture. Their biophysical characteristics can be easily adjusted, other external physical stimuli (such as cyclic stretch) can be applied, are used as bioinks for bioprinting, and can be used as vehicle facilitating MSCs engraftment and viability after transplantation [12,13,24]. This work reveals that L-HG is also an important source of bioactive proteins and derived vesicles that can modulate multiple cellular mechanisms. Specifically, we demonstrate that they can interact with alveolar epithelial cells facilitating lung repair. Interestingly, these results showed that the large amount of L-HG released vesicles have comparable therapeutic effects to those released by only MSCs when tested in a wound healing assay. Thus, the significance of these findings appears relevant since most studies carried out nowadays are not considering the presence of HG-derived vesicles (and their consequences) when hydrogels are used for cell culture.

Stem cell therapy has recently garnered much attention for treating respiratory diseases [25,26]. However, their therapeutic effects are minimal based on clinical trials [9,27,28]. The paracrine activity of MSCs, mediated partly by EVs, is being explored as a novel, promising approach to treating these diseases [29]. MSCs-derived EVs are generated and released into the extracellular environment to maintain tissue homeostasis. They are commonly characterized by a lipid membrane bilayer sharing some surface features [30]. However, their content, biogenesis, and release could depend on the surrounding environment [31]. As expected, here we found that MSCs derived from EVs can enhance wound healing in a model of lung repair. These findings support the notion that EVs could be used to treat severe ARDS patients that need an urgent and cell-free effective immunomodulatory treatment. In this work, MSCs were cultured in physiomimetic lung-derived hydrogels to understand how ECM components and MSCs could interact to improve the therapeutic capacity of their secretome. Surprisingly, the MSCs-derived EVs were a minor population among other ECM-vesicles released by the hydrogel, although results when normalized by the number of proteins showed that EVs have a greater effect in the cells than ECM-vesicles. In this way, MSCs-released particles showed to be much more specific for epithelial repair, confirming that ECM-vesicles present higher heterogeneity as expected.

It is known that hydrogels can contain subsets of EVs residing within the ECM [30,32,33]. Different types of vesicles have been very recently described from some decellularized tissues and scaffolds [17,32]. Although their role in repair and regeneration is still unknown, the scarce studies to date suggest that they could participate in modulating matrix remodeling [34] and tissue regeneration [33]. Interestingly, some of these ECM-vesicles seem to confer immunomodulatory effects on immune cells [35]. For instance, the application of matrix-bound vesicles on macrophages induces the expression of anti-inflammatory markers such as Fizz-1 and interleukin (IL)-4 indicating a polarization towards an M2 phenotype [30,35]. Accordingly, our results show that L-HG-derived vesicles also participate in tissue regeneration and repair based on a wound healing alveolar epithelial model. Our results could explain the protective role of intratracheal instillation of L-HG in a radiation-induced lung injury model [36], showing that L-HG could reduce epithelial–mesenchymal transition, inflammation, and oxidative damage induced by radiation [36]. In this context, further immunomodulatory studies are needed to assess the response generated by different cell types in the presence of the vesicles released by L-HG.

It should be also considered that hydrogels are degradable biomaterials. As previously reported [37], we observed a large amount of proteins released from the L-HG during the first 48 h after hydrogel gelation. These proteins are drastically reduced when a conventional EVs isolation protocol is applied (Figure 1). However, TEM images revealed the presence of small protein fragments from ECM degradation. Interestingly, we observed the presence of vesicles attached to these small ECM fragments confirming the existence of ECM-bound vesicles. These degradation products could confer some additional beneficial functions including antibacterial activity [19] and cell migration and proliferation [20]. However, the potential effect of these released products on alveolar epithelial wound healing needs to be studied in full detail in further studies.

## 5. Conclusions

The results obtained in this work provide novel insights into the therapeutic capacity of vesicles released from L-HG. These ECM-vesicles have the advantage of being present in the native lung, but their effects could differ from healthy or pathological conditions where ECM components are different (i.e., cystic fibrosis), opening new perspectives to better understand different diseases/conditions and their treatment. Moreover, the active role of ECM-vesicles should be considered specifically when tissue-derived hydrogels are used for cell culture. As shown in this work, the large amount of ECM-vesicles released to supernatants can mask the effects of the MSCs-derived EVs for lung repair. In some mechanistic studies, the impact of ECM-vesicles could be reduced by previously washing the hydrogel. A better knowledge of ECM-vesicles could be crucial to interpreting the results obtained from the secretome obtained from cells cultured on hydrogels. Further analysis of the different types of ECM-vesicles released from hydrogels and the identification of the functional proteins released from the ECM are still needed to better understand their role in other cell populations and conditions.

## Figures and Tables

**Figure 1 polymers-14-04907-f001:**
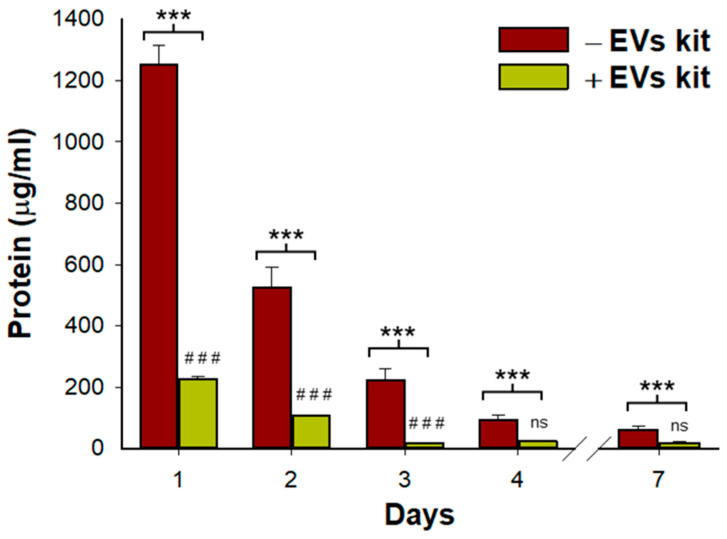
Total protein content and proteins released from bond-ECM vesicles measured in L-HG supernatants and after different daily washes. Whereas total protein was reduced every day, the portion of proteins from vesicles was reduced until day 3 reaching a plateau. (*n* = 3 per group). * Comparisons between total protein and proteins from ECM-vesicles and # represents comparisons between consecutive days in proteins from ECM-vesicles. *** and ### *p* < 0.001.

**Figure 2 polymers-14-04907-f002:**
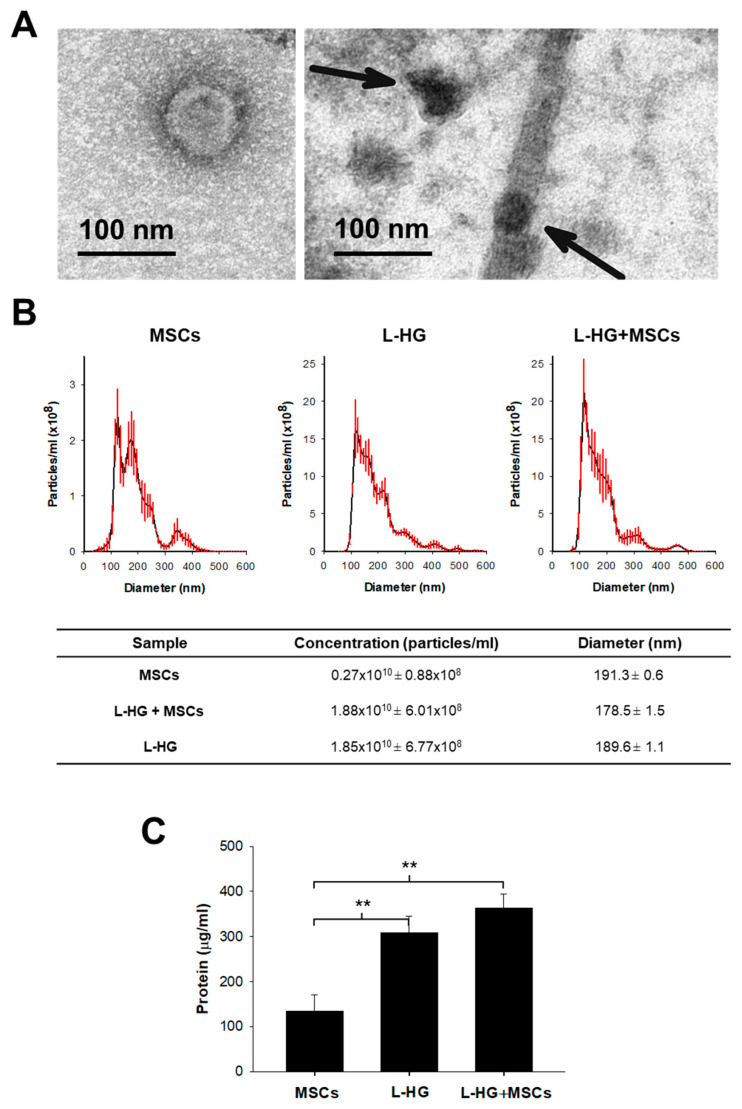
Microvesicle quantification and characterization in all groups. (**A**) TEM images showing an isolated vesicle (**left**) and ECM-bound vesicles (**right**). (**B**) NTA analysis revealed that the number of ECM-vesicles released from 3-days washed L-HG were still higher than EVs secreted by MSCs cultured in plastic. The NTA profile also shows that vesicles released from both L-HG and MSCs have a very similar profile in size distribution. (**C**) The protein content of the particles isolated in each condition confirmed the NTA quantification showing a 3-fold increase in L-HG with respect to MSCs. ** *p* < 0.01 (*n* = 5).

**Figure 3 polymers-14-04907-f003:**
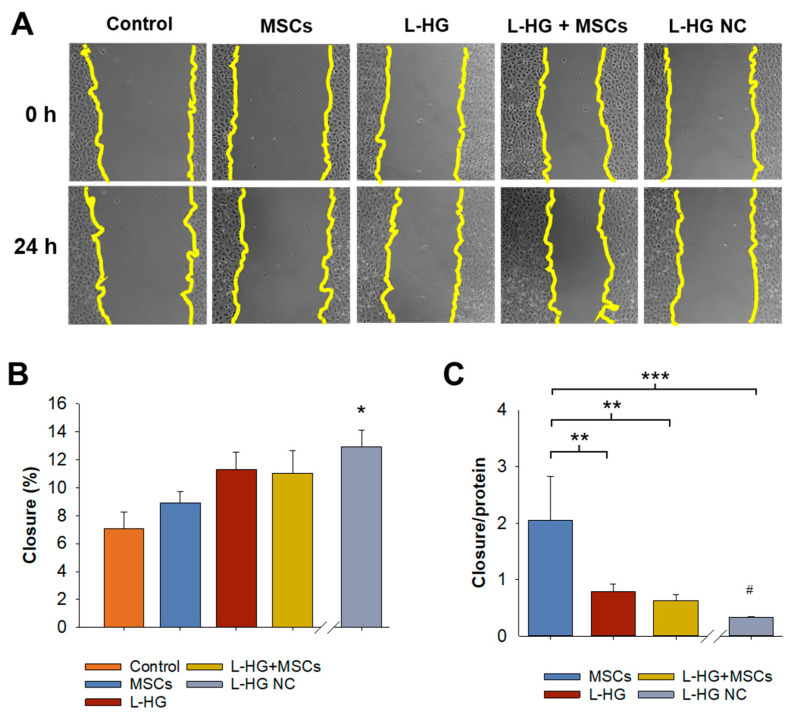
Functional characterization of EVs isolated from control, MSCs, pre-washed lung hydrogel (L-HG), L-HG with MSCs, and non-washed L-HG (L-HG NW). (**A**) Representative images of wound healing assay at 0 h and 24 h for each condition. (**B**) Wound closure was enhanced by applying MSCs, and L-HG-derived vesicles. (**C**) Wound closure normalized by total protein applied showed a higher effectivity of MSCs-derived EVs. * *p* < 0.05 and ** *p* < 0.01, *** *p* < 0.001 (*n* = 5). # Comparison between L-HG and L-HG NC, *p* < 0.05. Scale bar = 100 µm.

## Data Availability

Data supporting the findings of this study are available from the corresponding authors upon reasonable request.

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
