# Peer review of "Lung Extracellular Matrix Hydrogels-Derived Vesicles Contribute to Epithelial Lung Repair"

_polymers, 2022, doi:10.3390/polym14224907_

Round 1

Reviewer 1 Report

Comments:
The manuscript “Lung Extracellular Matrix Hydrogels – Derived Vesicles Contribute to Epithelial Lung Repair” by Isaac Almendros and colleagues studied the contribution of extracellular vesicles and ECM-vesicles released from mesenchymal stromal cells and lung-derived hydrogel (L-HG). They found the ECM-vesicles from L-HG can release 10-fold higher than EVs from conventional MSCs cells but have less therapeutic potential. However, the reviewer believes that additional points of clarifications could potentially be addressed to further strengthen the manuscript.

1.     What the lungs ECM-derived hydrogels looks like? I would recommend the author provide the SEM images or the photograph of the ECM-derived hydrogel.

2.     How was the stability of the ECM-derived hydrogel?

3.     Biodegradable is an important feature of biomaterials including hydrogel. Does this hydrogel degradable or not? I would recommend the author provide some experiments to show if this gel is safe to use.

4.     What is the cytotoxicity of the ECM-derived hydrogel? The author should provide the cytotoxicity experiment or some histology data to show that the hydrogel is safe to use in vivo.

5.     What is the loading amount or loading percentage of the ECM-bonded proteins? I would recommend the author include this data in the manuscript.

6.     In line 120, 130 and 132, please add a space before ‘°C’.

Author Response

What the lungs ECM-derived hydrogels looks like? I would recommend the author provide the SEM images or the photograph of the ECM-derived hydrogel.

Thank you for your suggestion. A representative SEM image of the ECM-derived hydrogel has been added as supplementary Figure 2A.

How was the stability of the ECM-derived hydrogel?

We are using hydrogels following a previous protocol with already characterized stability. Indeed, Pouliot and colleagues described the differences in the organization of the proteins network in the hydrogel after 14 days of gelation (R.A. Pouliot, et al. J. Biomed. Mater. Res. - Part A. 104 (2016) 1922–1935). This information has been added to the manuscript.

Biodegradable is an important feature of biomaterials including hydrogel. Does this hydrogel degradable or not? I would recommend the author provide some experiments to show if this gel is safe to use.

Thank you for your comment. The hydrogel is biodegradable. In fact, we observed it by the large amount of proteins released in the wash immediately after complete gelation (illustrated in Figure 1: with vs. without using EVs isolation protocol). Similarly, it has been described that a passive release of proteins can be collected. We observed that the highest protein release occurs within the first 48 h. Pouliot et al. observed a difference in the quantity and organization of proteins in the hydrogel network and in the density of non-integrated proteins from day 1 to day 14 (R.A. Pouliot, et al. J. Biomed. Mater. Res. - Part A. 104 (2016) 1922–1935). We have now included a paragraph about this relevant information in the discussion section, including the interesting information from the TEM images requested by Reviewer 2.

What is the cytotoxicity of the ECM-derived hydrogel? The author should provide the cytotoxicity experiment or some histology data to show that the hydrogel is safe to use in vivo.

We have previously described the biocompatibility of the L-HG (Falcones, et al. Polymers. 13 (2021). In fact, cells showed spread-like morphology indicative of their normal attachment and growth to L-HG. In the revised version, F-actin staining has been added to illustrate the attachment and growth of the cells (supplementary Figure 2).

What is the loading amount or loading percentage of the ECM-bonded proteins? I would recommend the author include this data in the manuscript.

The supplementary Table 1 provided shows the protein amount added in the alveolar epithelial wound-healing assay are added on the manuscript.

In line 120, 130 and 132, please add a space before ‘°C’.

Thanks. It has been corrected.

Reviewer 2 Report

The authors raised an interesting topic that EVs from ECMs could as well contribute to EVs final products if cells were cultured in the presence of ECM. However, the evidence could not fully support the claims that some fundamental characterization experiments were missing.

Results:

Electronic microscopy images & western blot are missing, and these two experiments are inevitable for any EV characterization.

The images of control groups in the wound healing experiments should be shown. Also, the results of the closure tests implied that the total amount of protein affected the recovery of epithelial cells. However, the recovery should be a result of specific proteins produced by HG rather than MSCs. Understanding or identifying the functional proteins is the key to EV studies. With these results, one can move further to observe the recovering/repairing results either in vitro or in vivo.

Also, the evidence of decellularization should be clearly shown in the context. No results of staining/sequencing/proteomics were given. The EVs from ECM-derived hydrogels would be suspected as residuals from the lungs.

Writing:

I would suggest the authors expand what you've done in this paper in the end introduction (~Line 87). The authors provided a solid and intriguing background intro, but what readers would like to know is your contribution to the field.

Some typos: Line 73, chemical; Line 210 Wound

Author Response

Electronic microscopy images & western blot are missing, and these two experiments are inevitable for any EV characterization.

TEM images have been added into the manuscript. Interestingly, after the EVs isolation protocol there are some remaining ECM fragments, and the images show the presence of vesicles bound to these fragments. Unfortunately, we do not have enough sample to characterize the proteins from the ECM fragments and EVs by western blot.

The images of control groups in the wound healing experiments should be shown. Also, the results of the closure tests implied that the total amount of protein affected the recovery of epithelial cells. However, the recovery should be a result of specific proteins produced by HG rather than MSCs. Understanding or identifying the functional proteins is the key to EV studies. With these results, one can move further to observe the recovering/repairing results either in vitro or in vivo.

Thank you very much for your comment. Control images of the wound healing experiments have been added in the revised manuscript. After the EVs isolation protocol, the amount of protein was reduced considerably but as shown by TEM images we cannot discard a potential effect of the remaining ECM fragments on wound closure. As commented previously, these images, in addition to the rest of results observed, reveal the difficulties of working with EVs when hydrogels are used for cell culture (a main message of this manuscript). We agree with the Reviewer in that the identification of the functional proteins is important, but we think this deserves a much more detailed new study including other tissue sources. We have included a last paragraph in the discussion section mentioning this issue and have changed the conclusion accordingly.

Also, the evidence of decellularization should be clearly shown in the context. No results of staining/sequencing/proteomics were given. The EVs from ECM-derived hydrogels would be suspected as residuals from the lungs.

We have now provided the amount of remaining DNA after lung decellularization and a supplementary Figure 1.

Writing:

I would suggest the authors expand what you've done in this paper in the end introduction (~Line 87). The authors provided a solid and intriguing background intro, but what readers would like to know is your contribution to the field.

Thank you. We have changed accordingly this part of the introduction.

Round 2

Reviewer 1 Report

Comments:

All the comments are addressed. Just one more question here. 

1. in Figure 2A, the TEM images have very dirty background and your scale bar is 100 um, which is too high. I would recommend the author provide one with clearer background also a zoomed in TEM images here. 

Author Response

in Figure 2A, the TEM images have very dirty background and your scale bar is 100 um, which is too high. I would recommend the author provide one with clearer background also a zoomed in TEM images here. 

Thank you for your comment, we apologize our mistake with the figure legend (100 µm). We have corrected in the image including the correct value (100 nm).

Reviewer 2 Report

The authors have addressed comments except for the western blot results. The scale bars of the TEM image presented in Figure 2A were 100um which I considered too large for EVs. Would that be possible if any other images with higher resolution and clean backgrounds can be provided?

Author Response

The authors have addressed comments except for the western blot results. The scale bars of the TEM image presented in Figure 2A were 100um which I considered too large for EVs. Would that be possible if any other images with higher resolution and clean backgrounds can be provided?

Thank you for your comment, we apologize our mistake with the figure legend (100 µm). We have corrected in the image including the correct value (100 nm).
